# Unplanned 30-Day Readmission in Glioblastoma Patients: Implications for the Extent of Resection and Adjuvant Therapy

**DOI:** 10.3390/cancers15153907

**Published:** 2023-08-01

**Authors:** Johannes Kasper, Johannes Wach, Martin Vychopen, Felix Arlt, Erdem Güresir, Tim Wende, Florian Wilhelmy

**Affiliations:** Department of Neurosurgery, University Hospital Leipzig, 04103 Leipzig, Germany; johannes.wach@medizin.uni-leipzig.de (J.W.); martin.vychopen@medizin.uni-leipzig.de (M.V.); felix.arlt@medizin.uni-leipzig.de (F.A.); erdem.gueresir@medizin.uni-leipzig.de (E.G.); tim.wende@medizin.uni-leipzig.de (T.W.); florian.wilhelmy@medizin.uni-leipzig.de (F.W.)

**Keywords:** glioblastoma, unplanned early readmission, extent of resection, radio-chemotherapy

## Abstract

**Simple Summary:**

Unplanned early hospital readmission (UER) within 30 days seems to be associated with reduced overall survival in patients diagnosed with glioblastoma (GBM). In this study, we retrospectively analyzed how the extent of tumor resection or adjuvant tumor treatment affected the prognosis in GBM patients that experienced UER. A total of 276 patients were included in the study. UER occurred in 13.4% of all cases and significantly reduced the median survival prognosis (5.7 vs. 14.5 months). Moreover, GBM patients suffering from UER did not benefit from extensive tumor resection (5.1 vs. 5.7 months). Concerning post-operative treatment strategies, the application of radio-chemotherapy prolonged the overall survival of GBM patients, even when UER occurred (1.1 (without post-operative therapy) vs. 4.3 (radiotherapy alone) vs. 7.8 months (radio-chemotherapy)). Therefore, GBM patients experiencing unplanned early readmission within 30 days still benefitted from more aggressive post-operative therapy.

**Abstract:**

Background: Unplanned early readmission (UER) within 30 days after hospital release is a negative prognostic marker for patients diagnosed with glioblastoma (GBM). This work analyzes the impact of UER on the effects of standard therapy modalities for GBM patients, including the extent of resection (EOR) and adjuvant therapy regimen. Methods: Records were searched for patients with newly diagnosed GBM between 2014 and 2020 who were treated at our facility. Exclusion criteria were being aged below 18 years or missing data. An overall survival (OS) analysis (Kaplan–Meier estimate; Cox regression) was performed on various GBM patient sub-cohorts. Results: A total of 276 patients were included in the study. UER occurred in 13.4% (*n* = 37) of all cases, significantly reduced median OS (5.7 vs. 14.5 months, *p* < 0.001 by logrank), and was associated with an increased hazard of mortality (hazard ratio 3.875, *p* < 0.001) in multivariate Cox regression when other clinical parameters were applied as confounders. The Kaplan–Meier analysis also showed that patients experiencing UER still benefitted from adjuvant radio-chemotherapy when compared to radiotherapy or no adjuvant therapy (*p* < 0.001 by logrank). A higher EOR did not improve OS in GBM patients with UER (*p* = 0.659). Conclusion: UER is negatively associated with survival in GBM patients. In contrast to EOR, adjuvant radio-chemotherapy was beneficial, even after UER.

## 1. Introduction

Glioblastoma (GBM) is the most common brain-derived tumor. Characterized by high mitotic activity, aggressive invasive behavior, central necrosis, and neo-angiogenesis, it is classified as WHO grade 4 [1]. It is more common in men, and the median age at diagnosis is 65 years [2]. The standard therapy is maximum safe resection aiming for gross-total resection, followed by adjuvant radio-chemotherapy (up to 60.0 gray and concomitant application of temozolomide), and six cycles of maintenance chemotherapy with temozolomide [3]. Despite intense research efforts and newly available treatment options, such as tumor-treating fields [4] or a new systemic therapy regimen [5], the overall survival rate remains poor. Several clinical and radiological parameters were identified as being associated with survival prognosis. This includes the extent of resection (EOR), O6-methylguanine DNA methyltransferase (MGMT) promoter methylation status [6], patient age at time of diagnosis [7], tumor location, or occurrence of neurological deficits [8,9]. 

Unplanned early readmission (UER) in general represents a significant economic burden on health care systems [10,11] and might be used as a marker for in-hospital quality of care [12,13]. Concerning neurosurgical cohorts, several works reported an association between UER and a shortened overall survival prognosis, e.g., in glioblastoma [14,15,16]. Moreover, previous data indicated that UER can be associated with preventable adverse events, resulting in reduced quality of care for these patients [12,17]. Due to the affection of the central nervous system, inpatient treatment of early readmitted neurosurgical patients is commonly indispensable [12].

While the negative association of UER with the survival prognosis of GBM patients has been reported before, the impact of unplanned early readmission on standard treatment modalities for glioblastoma patients remains unclear. Therefore, we evaluated the influence of unplanned early readmission on patients suffering from GBM, with a focus on the extent of resection and adjuvant therapy regimen. 

## 2. Methods

### 2.1. Patient Selection and Treatment

Data collection and analysis were approved by the ethical committee of the Medical Faculty, University of Leipzig, and carried out in accordance with data protection guidelines. Informed consent for retrospective data analysis was obtained from all patients treated in the Neurosurgical Department of Leipzig University. The medical records were checked for all patients with first diagnosis of IDH-wildtype glioblastoma between 1 January 2014 and 31 December 2020 treated at Leipzig University Hospital. Patients aged below 18 years or missing clinical or pathological data were excluded. All tumor cases in our department and neurooncological center were discussed in a weekly, interdisciplinary tumor board, and therapy regimens were determined based on the current treatment guidelines for glioma therapy.

### 2.2. Clinical, Pathological, and Radiological Assessment

Medical records were analyzed for age at date of diagnosis, sex, peri-operative clinical performance, main onset symptoms, length of stay (LOS) at the intensive care (ICU) unit or our department in general, occurrence of post-operative complications, unplanned early readmission (UER), discharge disposition, and adjuvant therapy regimen. The date of diagnosis was set as the date of neurosurgery with neuropathological proof of glioblastoma. 

UER was defined as unexpected readmission to any hospital that required inpatient treatment within 30 days after initial release from our facility. Post-operative complications were defined as every medical condition that occurred after initial neurosurgical tumor extirpation or biopsy with temporary or permanent decrease in the neurological or physical status of patients.

The medical research council neurological performance scale (MRC-NPS) adjusted by Bleehen et al. [18] was used to assess neurological performance with (1) no neurological deficit; (2) some neurological deficit but function adequate for useful work; (3) neurological deficit causing moderate functional impairment (difficulty to move limbs, moderate dysphasia, moderate paresis, and some visual disturbance); (4) neurological deficit causing major functional impairment (inability to move limbs and gross speech or visual disturbances); and (5) inability to make conscious responses. The difference in MRC-NPS was calculated as post-operative values minus pre-operative values. Hence, positive values indicate neurological deterioration, and vice versa.

Histopathological diagnosis and immunohistochemical status were extracted from neuropathology reports. IDH mutation status and MGMT promoter methylation of all GBM samples were determined using immunohistochemistry and pyrosequencing, or nucleic acid amplification followed by pyrosequencing. 

Overall survival (OS) was defined as the time between date of neurosurgery and date of death. The date of death, if not provided by our hospital database, was collected from the Leipzig Cancer Registry. Dates were assessed on 31 December 2022. If death did not occur by then or if patients were lost to follow-up, the date of last contact with our department was integrated into statistical analysis as censored value.

Extent of resection (EOR) was retrospectively determined by revising MRI T1 sequences with and without contrast. Analysis was carried out employing iPlan Cranial software (version 3.0.5, Brainlab AG, Munich, Germany). If burr hole trepanation with needle biopsy was performed or the extent of resection was unknown due to missing post-operative MR imaging, EOR was set at 0%. 

### 2.3. Statistical Analysis

Statistical analysis was performed for the entire cohort and sub-cohorts (patients with or without UER) using SPSS statistics software version 29.0.0.0 (IBM, Armonk, NY, USA). Comparative sub-cohort analysis was carried out with Mann–Whitney U test. Survival analysis was performed using Kaplan–Meier estimate as well as Cox regression for univariate and multivariate calculations of survival probability. Survival rates from Kaplan–Meier analysis were tested for statistical significance via logrank. *p*-values below 0.05 were considered statistically significant. Hazard ratios (HR) from Cox regression are provided with 95% confidence intervals (95 CI) and were considered statistically significant if 1 was excluded by 95 CI. In order to analyze the impact of UER on different treatment regimens, a cutoff for EOR was calculated. Therefore, time-dependent receiver operator characteristic (ROC) analysis was performed, and the optimal cutoff point was defined as the value that maximizes the Youden index (parameter value for which sensitivity + specificity − 1 is maximal). After EOR was categorized according to cutoff values, a second univariate analysis was carried out with and Kaplan–Meier estimate, as stated above.

## 3. Results

### 3.1. Baseline Data

Between 2014 and 2021, 294 patients with newly diagnosed GBM were treated at our facility, of whom 276 were included in this study. In accordance with larger databases, male patients were more often diagnosed with glioblastoma, and the median age at diagnosis was 68.8 years (Table 1). Bilateral hemispheric tumor location (27.0% vs. 12.6%, *p* = 0.033), post-operative neurological deterioration (48.6% vs. 24.7%, *p* = 0.009), and post-operative complications (51.4% vs. 31.4%, *p* = 0.012) were significantly more frequent within the UER sub-cohort. Concerning treatment strategies, EOR (*p* = 0.111) and adjuvant therapy regimens were equally distributed among the groups (*p* = 0.36).

### 3.2. Unplanned Early Readmission

UER occurred in 37 cases (13.4% of the entire cohort). Progressive neurological deterioration was the most frequent cause for UER (24 of 37 cases, 64.9%). Among those, five patients suffered from early rapid tumor progression (13.5%), five patients were readmitted due to epileptic seizures, two patients developed hydrocephalus (5.4%), and two patients suffered from intracranial hemorrhage. Six patients (16.2%) developed infectious complications, with the diagnosis of surgical site infection (SSI) in three cases (8.1%). Two patients (5.4%) were readmitted due to insufficient home care. Other medical reasons for UER included one case (2.7%) of urethral hemorrhage, intestinal ischemia, coprostasis, hypertensive crisis, and suicidal attempt, respectively. Surgical treatment was performed for SSI, hydrocephalus, and in one case of intracranial hemorrhage (16.2% of all cases with UER).

### 3.3. Overall Survival

Kaplan–Meier analysis of GBM patients with or without UER is presented in Figure 1. The median overall survival of patients suffering from UER was significantly shorter (5.7 vs. 14.5 months, *p* < 0.001 by logrank). 

Univariate Cox regression revealed significant influences for all the included parameters except the difference in MRC-NPS and the occurrence of post-operative complications (Table 2). When adjusted as confounders in a multivariate analysis, UER (hazard ratio (HR) 3.875, *p* < 0.001) and discharge disposition other than home (HR 1.671, *p* < 0.001) were significantly associated with an increased hazard of mortality. Peri-operative increase or stability of neurological performance calculated via the difference in MRC-NPS (HR 0.556, *p* = 0.042), higher extent of resection (HR 0.995, *p* = 0.025), more aggressive adjuvant therapy (HR 0.353, *p* < 0.001), and positive MGMT promoter methylation status (HR 0.594, *p* = 0.002) were significantly associated with a reduction in mortality risk (Table 2).

### 3.4. Readmission, Extent of Resection, and Adjuvant Therapy

The optimal cutoff of EOR for the entire cohort was 90.58% (AUC: 0.63). Survival graphs of sub-cohorts calculated via the Kaplan–Meier estimate are presented in Figure 2. UER significantly reduced the median overall survival for patients receiving concomitant radio-chemotherapy (7.8 (UER) vs. 20.4 months (without UER)), as well as radiotherapy alone (4.3 (UER) vs. 7.9 months (without UER), both *p* < 0.001), but an analysis of the UER sub-cohort alone revealed beneficial effects on survival prognosis when adjuvant radio-chemotherapy was applied (1.1 (without therapy) vs. 4.3 (Rx) vs. 7.8 months (RCx), *p* < 0.001). In the UER sub-group, EOR yielded no survival benefit (5.1 (EOR below cutoff) vs. 5.7 months (EOR above cutoff), *p* = 0.659). The median OS of patients without UER was significantly increased when an EOR above the cutoff was achieved (9.7 (EOR below cutoff) vs. 20.9 (EOR above cutoff) months, *p* < 0.001).

## 4. Discussion

The thirty-day readmission rate for inpatient care is an important financial benchmark, considered a marker for quality of care and associated with high costs for health care systems [10,19,20]. Several studies have also shown that UER is associated with a reduced outcome for various medical conditions, especially when survival rates are analyzed [21,22,23,24,25]. In neurosurgery, the most frequent causes for unplanned readmission include post-operative infections and medical complications, such as thromboembolic events, seizures, intracranial hemorrhage, and shunt or ventricle catheter-associated complications [14,15,26], especially in pediatric neurosurgery [27,28]. Neurological complications accounted for the majority of readmission cases in our cohort, which is in line with previous data investigating UER in glioblastoma cohorts [14,15]. The overall readmission rates vary between 6.0% and 21.0% and are highly dependent on the neurosurgical index diagnosis [17,26,29]. Thirty-day readmission rates for patients suffering from glioblastoma were reported before at 7.5% [15] and 20.3% [30], in comparison with 13.8% in our cohort. Therefore, GBM patients are at higher risk of experiencing UER when compared to other neurosurgical sub-cohorts [26], including patients treated at our Neurosurgery Department [17]. The reported factors associated with early readmission in GBM patients were reduced overall physical performance determined via KPS, discharge disposition other than home, MGMT methylation status, location of tumor in eloquent brain areas, or intra-operative hypertension [15,16,30,31]. We also found higher frequencies of bi-hemispheric tumor location, peri-operative neurological deterioration, and the occurrence of post-operative complications in early readmitted glioblastoma patients. In accordance to earlier data [14,15], UER had a significant impact on GBM patient prognosis, with a reduction in the median overall survival of over 50% in the cohort reported here. Early readmission was also independently associated with a higher hazard of mortality among other well-established prognostic factors when multivariate Cox regression analysis was performed. In addition, the study presented here revealed a significant impact of UER on the prognostic effects of standard treatment modalities. As overall survival was reduced in general, patients who experienced early readmission strongly benefited from adjuvant radiotherapy with concomitant chemotherapy when compared to radiotherapy alone or no adjuvant therapy. When analyzed for the extent of resection, survival analysis showed no significant prognostic effect of higher EOR rates for patients in the UER sub-cohort. The beneficial effects of a more aggressive tumor resection were abrogated by the occurrence of early readmission. The underlying mechanism remains unclear. As EOR rates were equally distributed among sub-cohorts, co-dependencies between UER and other prognostic factors that were not considered for this study cannot be ruled out. A higher EOR might be associated with an increased risk of reduced peri-operative neurological performance, which was found more often in the UER sub-cohort and was also the most frequent cause leading to unplanned early readmission. Multivariate analysis, on the other hand, revealed a significant association between shortened overall survival and the occurrence of early readmission, independently from other well-established parameters (including reduced neurological performance). Also, patients in the UER subgroup suffered more often from bilateral glioblastoma, which might have reduced their eligibility for gross-total resection in retrospective, therefore showing no impact of the extent of resection on survival.

The study presented here is limited by its retrospective design. Selection bias cannot be fully ruled out. Based on the data presented here, no conclusions concerning the direct association or causality of clinical factors or treatment modalities with UER can be drawn. It remains, therefore, unclear how to avoid UER in GBM patients. Moreover, corticoid therapy was not considered for data analysis. Neurological deterioration might be induced by reducing corticoid therapy too rapidly, leading to UER. Due to the small number of non-tumor-related cases of UER, non-neurosurgical side-morbidities were not considered in detail for this work.

Unplanned early readmission remains a considerable marker for quality of care but also for financial aspects of health care systems. For patients with a very limited prognosis, including glioblastoma, UER should be avoided whenever possible. It is important for healthcare providers to identify patients at risk of unplanned early readmission in order to address the above-mentioned issues. Future studies should therefore explore factors leading to UER in GBM patients as well as the effects of recurrent surgery on UER. 

## 5. Conclusions

Unplanned early readmission was strongly associated with reduced overall survival in our cohort. When stratified for standard therapy modalities and in contrast to standard glioblastoma cohorts, an increased extent of resection yielded no benefit on survival prognosis in patients experiencing unplanned early readmission. However, radio-chemotherapy was still highly beneficial, even after an unplanned early readmission. 

## Figures and Tables

**Figure 1 cancers-15-03907-f001:**
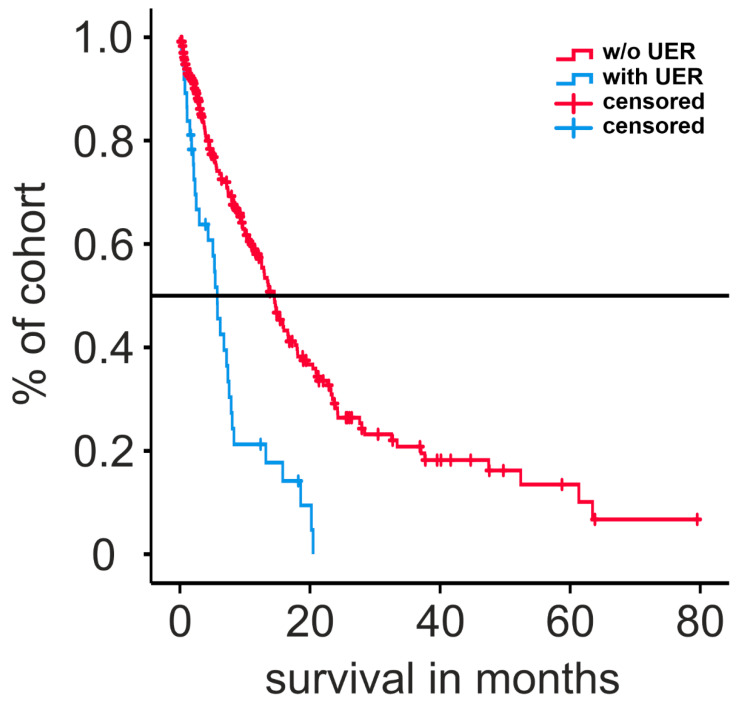
Survival graphs of patients with or without unplanned early readmission calculated with Kaplan–Meier estimate. UER: unplanned early readmission.

**Figure 2 cancers-15-03907-f002:**
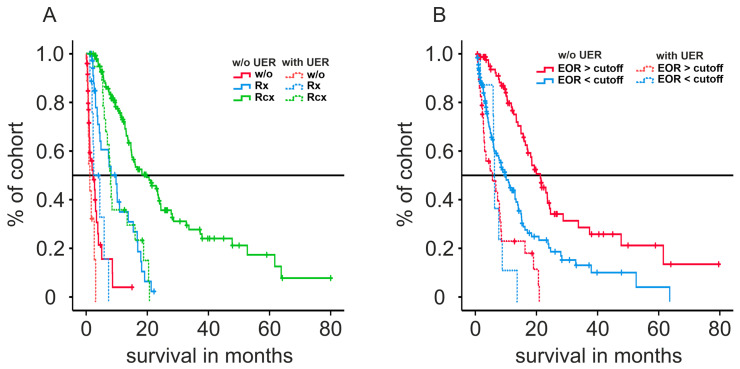
Survival graphs of patients with or without unplanned early readmission calculated with Kaplan–Meier estimate after stratification for adjuvant therapy regimen (**A**) or extent of resection (**B**). EOR: extent of resection; RCx: radio-chemotherapy; Rx: radiotherapy; UER: unplanned early readmission.

**Table 1 cancers-15-03907-t001:** Baseline data of the entire cohort and sub-cohorts sorted by occurrence of unplanned early readmission.

		All	without UER	with UER	*p* Value
*n*		276	239	37	N/A
Sex	male	175 (63.4)	150 (62.8)	25 (67.6)	0.573
	female	101 (36.6)	89 (37.2)	12 (32.4)	
Age, years		68.4, 32.6–86.7	69.9, 32.6–86.7	66.5, 33.8–84.4	0.065
Hemisphere	right	123 (44.6)	111 (46.4)	12 (32.4)	0.033 *
	left	113 (40.9)	98 (41.0)	15 (40.5)	
	bilateral	40 (14.5)	30 (12.6)	10 (27.0)	
Lobe	frontal	59 (21.4)	54 (22.6)	4 (10.8)	0.229
	parietal	46 (16.7)	40 (16.7)	6 (16.2)	
	temporal	76 (27.5)	64 (26.8)	12 (32.4)	
	occipital	14 (5.1)	14 (5.9)	0	
	multilocular	76 (27.5)	62 (25.9)	14 (37.8)	
	other	5 (1.8)	5 (2.1)	0	
KPS		80, 20–100	80, 20–100	80, 40–100	0.782
MRC-NPS	pre-operative	2, 1–5	2, 1–5	2, 1–5	0.186
	post-operative	3, 1–5	3, 1–5	3, 1–5	0.443
Difference in MRC-NPS ^†^	<0	27 (9.8)	25 (10.5)	2 (5.4)	0.009 *
	=0	172 (62.3)	155 (64.9)	17 (45.9)	
	>0	77 (27.9)	59 (24.7)	18 (48.6)	
Main onset symptom	seizure	43 (15.6)	37 (15.5)	6 (16.2)	0.148
	motor deficit	78 (28.3)	74 (31.0)	4 (10.8)	
	aphasia/dysphasia	44 (15.9)	35 (14.6)	9 (24.3)	
	visual deficit	11 (4.0)	9 (3.8)	2 (5.4)	
	cognitive deficit	21 (7.6)	19 (7.9)	2 (5.4)	
	change in character	22 (8.0)	19 (7.9)	3 (8.1)	
	headache	30 (10.7)	22 (9.2)	8 (21.6)	
	reduced vigilance	6 (2.2)	6 (2.5)	0	
	none	5 (1.8)	4 (1.7)	1 (2.7)	
	other	16 (5.8)	14 (5.9)	2 (5.4)	
EOR, %		72.2, 0–100	74.5, 0–100	36.7, 0–100	0.111
Post-operative complication	without	182 (65.9)	164 (68.6)	18 (48.6)	0.012 *
	with	94 (34.1)	75 (31.4)	19 (51.4)	
LOS, days		14, 4–72	14, 4–72	14, 6–37	0.705
LOS on ICU, days		1, 1–43	1, 1–43	1, 1–29	0.109
MGMT status	negative	123 (44.6)	108 (45.2)	15 (40.5)	0.580
	positive	153 (55.4)	131 (54.8)	22 (59.5)	
Adjuvant therapy	without	63 (22.8)	54 (22.6)	9 (24.3)	0.36
	radiotherapy	48 (17.4)	39 (16.3)	9 (24.3)	
	radio-chemotherapy	165 (59.8)	146 (61.1)	19 (51.4)	
Discharge disposition	home	202 (73.2)	176 (73.6)	26 (70.2)	0.776
	radio-oncology	32 (11.6)	27 (11.3)	4 (10.8)	
	rehabilitation clinic	17 (6.2)	12 (5.0)	5 (13.5)	
	palliative care/hospice	25 (9.1)	24 (10.0)	2 (5.4)	

For continuous and ordinal variables, the median with corresponding range is shown. For categorial variables, percent of corresponding n is shown in brackets. *p* values were calculated by Mann–Whitney U test comparing sub-cohort distribution of analyzed clinical factors. EOR: extent of resection; ICU: intensive care unit; N/A: not applicable; KPS: Karnofsky Performance Scale; LOS: length of stay; MRC-NPS: medical research council neurological performance scale; UER: unplanned early readmission. * Statistical significance. ^†^ Post-operative NPS minus pre-operative NPS (positive values indicate neurological deterioration and vice versa).

**Table 2 cancers-15-03907-t002:** Survival probability calculated via univariate and multivariate Cox regression.

				Multivariate Cox Regression		
	HR	95 CI	*p* Value	HR	95 CI	*p* Value
UER	2.755	1.852–4.098	<0.001 *	3.875	2.473–6.072	<0.001 *
Age	1.028	1.012–1.044	<0.001 *	1.013	0.995–1.031	0.156
Hemisphere	1.331	1.06–1.673	0.014 *	1.244	0.968–1.599	0.089
Location	1.282	1.151–1.427	<0.001 *	1.103	0.974–1.249	0.122
KPS	0.982	0.974–0.989	<0.001 *	0.997	0.981–1.013	0.707
MRC-NPS, pre-operative	1.459	1.238–1.719	<0.001 *	0.887	0.557–1.414	0.615
MRC-NPS, post-operative	1.621	1.368–1.921	<0.001 *	1.473	0.991–2.189	0.056
MRC-NPS, difference ^†^	1.178	0.908–1.527	0.217	0.556	0.316–0.978	0.042 *
Main onset symptom	1.086	1.030–1.145	0.002 *	1.045	0.98–1.114	0.181
EOR	0.992	0.989–0.995	<0.001 *	0.995	0.992–0.999	0.025 *
Post-operative complication	1.380	0.999–1.906	0.051	0.987	0.64–1.523	0.953
LOS	1.033	1.015–1.052	<0.001 *	0.975	0.946–1.004	0.094
LOS on ICU	1.076	1.048–1.104	<0.001 *	1.01	0.971–1.05	0.63
MGMT status	0.591	0.432–0.809	0.001 *	0.594	0.426–0.829	0.002 *
Adjuvant therapy	0.281	0.225–0.351	<0.001 *	0.353	0.266–0.468	<0.001 *
Discharge disposition	2.098	1.776–2.479	<0.001 *	1.671	1.359–2.054	<0.001 *

95 CI: 95% confidence interval; EOR: extent of resection; HR: hazard ratio; ICU: intensive care unit; KPS: Karnofsky Performance Scale; LOS: length of stay; MRC-NPS: medical research council neurological performance scale; UER: unplanned early readmission. * Statistical significance. ^†^ Post-operative NPS minus pre-operative NPS.

## Data Availability

Data of this work are available from corresponding author upon reasonable request.

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
