# Peer review of "Unplanned 30-Day Readmission in Glioblastoma Patients: Implications for the Extent of Resection and Adjuvant Therapy"

_cancers, 2023, doi:10.3390/cancers15153907_

Round 1

Reviewer 1 Report

The authors analysed a very interesting neurooncological aspect of early clinical readmission after GBM surgery. To my opinion there are only few minor remarks-

1. What about preoperative corticoid medication? Corticoids are given preoperative in most clinics and fast reduction may also be a problem.

2. please refer to side-morbidities of the patients. How many patients have known for example cardiac problems? Dis this affect UER?

3. please discuss the impact of these results in your clinical daily routine? Is there a way to reduce UER?

4. a comparsion to patients after  recurrent surgery would be interesting.

Author Response

We thank the reviewer for his:her remarks and the positive feedback. It helped to improve the manuscript, especially the limitations paragraph.

  1. What about preoperative corticoid medication? Corticoids are given preoperative in most clinics and fast reduction may also be a problem.

Indeed, the (sub-)cohorts where not analyzed for corticoids but it is an important clinical aspect to consider. A short notice has been added to our limitations.

  1. please refer to side-morbidities of the patients. How many patients have known for example cardiac problems? Dis this affect UER?

The reviewer is correct, side-morbidities were not especially considered by our analysis. The majority of UER cases occurred due to tumor-related complications. Only 8 cases were treated outside of our neurosurgery department (3 cases of infections other than SSI and 5 cases due to various conditions). Considering this small number, larger study cohorts would be needed to further explore the effect of non-neurosurgical morbidities. A short note to limitations was added considering your remark.

  1. please discuss the impact of these results in your clinical daily routine? Is there a way to reduce UER?

Due to retrospective study design, no direct associations or causalities can be identified between clinical factors or therapy modalities with UER. Hence, an answer of how to avoid future cases of UER cannot be provided based on our data. This is indeed an important limitation of our study which is therefore added within the limitation subsection of the revised manuscript.

  1. a comparsion to patients after  recurrent surgery would be interesting.

We agree. A note was added at the end of the discussion considering further analysis after recurrent surgery.

Reviewer 2 Report

Dear Authors,

1.  Provide a background that contains the scientific rationale for the study. The introduction should end with a statement of the aims of the study. Each statement of fact should be supported by references.

2. Clearly describe the selection of observational or experimental subjects, inclusion and exclusion criteria, randomization, and the use of controls.

3 Clearly describe the selection of observational or experimental subjects, inclusion and exclusion criteria, randomization, and the use of controls. Please note that for a clinical research study, the methods should be described in sufficient detail so that the reader can reproduce the study

4 Include details of the power analysis used to determine the study size.”

5 The data acquisition protocol, procedures, investigated parameters, methods of measurements, and apparatus should be described in sufficient detail to allow other scientists to reproduce the results.

6 Sources for all reagents and equipment should be given (Company, City, Country).

7 The statistical methods should be described in detail to enable verification of the reported results.

8 Demographic and clinical data, must be provided for clinical research.

9 The results should be presented in a logical sequence in the text, tables, and illustrations. Restrict tables and figures to the number needed to explain and support the results. Do not duplicate data in graphs and tables.

10 Provide statistical support for 'significant' results (such as a adjusted p-value).

Discussion

11 This section should be organized clearly with concise paragraphs, beginning with a summary and interpretation of the study findings. Do not duplicate the content of the main Background or Results sections in the Discussion section.”

12 The findings from the study should be compared with findings from previously published studies. Discuss the implications of the findings and suggest future studies.

13 A mandatory requirement for all original Research Articles is the inclusion of a paragraph at the end of the Discussion section that clearly describes the limitations of the study. The purpose of this is to show insight into the limitations of the study methods and data analysis and interpretation.

Conclusions

14 A concise and clear short paragraph should match the conclusions given at the end of the Abstract.

15 Avoid giving statements and conclusions that are not completely supported by the results of the study.

16  Include details of the power analysis used to determine the study size.

17 The data acquisition protocol, procedures, investigated parameters, methods of measurements, and apparatus should be described in sufficient detail to allow other scientists to reproduce the results.

18  Sources for all reagents and equipment should be given (Company, City, Country).”

19. The statistical methods should be described in detail to enable verification of the reported results.

20 Please include a maximum of 5 Keywords using terms from the Medical Subject Heading (MeSH) database on the PubMed site (http://www.ncbi.nlm.nih.gov/mesh).”

21 Please cite and introduce similar studies to your study so that you can discuss and compare previous findings with your data in the Discussion section. 

22 Please divide the main Methods section into subsections with clear subsection headings.

23 Please begin the main Methods section with the Ethical statement that includes the Ethics Committee approval details and also the details of patient informed consent.

24 Please divide the main Results section into subsections with clear subsection headings.
25 Each Table should have a clear and descriptive title. All abbreviations should be defined in a footnote for each Table, figure.

26 The graphical abstract is a good idea.

27. More data to support findings are required.

28 Please clearly indicate if it was a retrospective or prospective study.

29. Please submit the paper with the track changes option and in the rebuttal letter clearly state the lines with changes and write if you mentioned a clear or with track changes option.

Major revision is required.

Author Response

We thank the reviewer for his:her remarks that helped to improve our manuscript. A detailed response is given below:

  1. Provide a background that contains the scientific rationale for the study. The introduction should end with a statement of the aims of the study. Each statement of fact should be supported by references.

The scientific background of our study is given with the introduction. References are provided as shown in the manuscript. Within the introduction’s last paragraph, the study aim is defined (lines 56-60).

  1. Clearly describe the selection of observational or experimental subjects, inclusion and exclusion criteria, randomization, and the use of controls.

The first subsection of ‘methods’ provides information concerning data acquisition as well as inclusion/exclusion criteria.

3 Clearly describe the selection of observational or experimental subjects, inclusion and exclusion criteria, randomization, and the use of controls. Please note that for a clinical research study, the methods should be described in sufficient detail so that the reader can reproduce the study

 See our answer to remark no 2.

4 Include details of the power analysis used to determine the study size.”

Due to our study’s retrospective nature, a power analysis to define a cohort size is not necessary.

5 The data acquisition protocol, procedures, investigated parameters, methods of measurements, and apparatus should be described in sufficient detail to allow other scientists to reproduce the results.

Please refer to methods, 2nd subsection.

6 Sources for all reagents and equipment should be given (Company, City, Country).

Reagents or special equipment were not used for our study.

7 The statistical methods should be described in detail to enable verification of the reported results.

Please refer to methods, 3rd subsection.

8 Demographic and clinical data, must be provided for clinical research.

Please refer to results, 1st subsection.

9 The results should be presented in a logical sequence in the text, tables, and illustrations. Restrict tables and figures to the number needed to explain and support the results. Do not duplicate data in graphs and tables.

In our opinion, results as well as tables are presented according to the above mentioned requirements.

10 Provide statistical support for 'significant' results (such as a adjusted p-value).

All statistical results are provide with p values to define statistical significance.

Discussion

11 This section should be organized clearly with concise paragraphs, beginning with a summary and interpretation of the study findings. Do not duplicate the content of the main Background or Results sections in the Discussion section.”

In our opinion, the mentioned section is provided as stated above.

12 The findings from the study should be compared with findings from previously published studies. Discuss the implications of the findings and suggest future studies.

In our opinion, the mentioned section is provided as stated above.

13 A mandatory requirement for all original Research Articles is the inclusion of a paragraph at the end of the Discussion section that clearly describes the limitations of the study. The purpose of this is to show insight into the limitations of the study methods and data analysis and interpretation.

Limitations are provided, as needed, at the end of the manuscript’s discussion.

Conclusions

14 A concise and clear short paragraph should match the conclusions given at the end of the Abstract.

In our opinion, this part is provided as mentioned above.

15 Avoid giving statements and conclusions that are not completely supported by the results of the study.

In our opinion, this part is provided as mentioned above.

16  Include details of the power analysis used to determine the study size.

Please refer to our answer to remark number 4.

17 The data acquisition protocol, procedures, investigated parameters, methods of measurements, and apparatus should be described in sufficient detail to allow other scientists to reproduce the results.

Please refer to our answer to remark number 5.

18  Sources for all reagents and equipment should be given (Company, City, Country).”

Please refer to our answer to remark number 6.

  1. The statistical methods should be described in detail to enable verification of the reported results.

Please refer to our answer to remark number 7.

20 Please include a maximum of 5 Keywords using terms from the Medical Subject Heading (MeSH) database on the PubMed site (http://www.ncbi.nlm.nih.gov/mesh).”

Keywords are provided at the bottom of page 1.

21 Please cite and introduce similar studies to your study so that you can discuss and compare previous findings with your data in the Discussion section. 

Please refer to our answer to remark number 12.

22 Please divide the main Methods section into subsections with clear subsection headings.

The mentioned section is divided into three subsections with subsection headings.

23 Please begin the main Methods section with the Ethical statement that includes the Ethics Committee approval details and also the details of patient informed consent.

The required information is given within the first two sentences of the 1st subsection of ‘Methods’.

24 Please divide the main Results section into subsections with clear subsection headings.

The mentioned section is divided into four subsection with subsection headings.

25 Each Table should have a clear and descriptive title. All abbreviations should be defined in a footnote for each Table, figure.

In our opinion, tables are provided in the required manner.

26 The graphical abstract is a good idea.

A graphical abstract may not suit the retrospective study design and abstract findings of our study.

  1. More data to support findings are required.

We agree. The need for future studies to explore the here described effects of our work and to address limitations of our study is expressed at the end of the discussion.

28 Please clearly indicate if it was a retrospective or prospective study.

The retrospective study design is clearly stated within simple summary (line 11), methods (line 65) and discussion of limitations (line 234)

  1. Please submit the paper with the track changes option and in the rebuttal letter clearly state the lines with changes and write if you mentioned a clear or with track changes option.

The revised manuscript was edited with track options.

Reviewer 3 Report

Authors analyzed Unplanned early readmission (UER) within 30 days after hospital release, which is 20 a negative prognostic marker for patients diagnosed with glioblastoma (GBM). They showed UER is negatively associated with survival in GBM patients. In contrast to EOR, adjuvant ra-33 dio-chemotherapy was beneficial even after UER.

Author Response

Authors analyzed Unplanned early readmission (UER) within 30 days after hospital release, which is 20 a negative prognostic marker for patients diagnosed with glioblastoma (GBM). They showed UER is negatively associated with survival in GBM patients. In contrast to EOR, adjuvant ra-33 dio-chemotherapy was beneficial even after UER.

We thank the reviewer for his:her positive feedback.

Round 2

Reviewer 2 Report

None

None